METHODS AND RESOURCES

# Extreme diversity of phage amplification rates and phage–antibiotic interactions revealed by PHORCE

Yuval Mulla[1,2]*[◉], Janina Müller[1][◉], Denny Trimcev[1], Tobias Bollenbach[1,3]*

**1** Institute for Biological Physics, University of Cologne, Cologne, Germany, **2** Molecular Microbiology, A-LIFE, AIMMS, Vrije Universiteit, Amsterdam, The Netherlands, **3** Center for Data and Simulation Science, University of Cologne, Cologne, Germany

◉ These authors contributed equally.
* y.mulla@vu.nl (YM); t.bollenbach@uni-koeln.de (TB)

## Abstract

Growth rate plays a fundamental role in microbiology and serves as an important proxy for fitness in evolution. While high-throughput measurements of bacterial growth rates are easily performed in any microbiology laboratory, similar methods are lacking for bacteriophages. This gap hinders systematic comparisons of important phage phenotypes, such as their amplification rate in bacterial populations and their bactericidal effect, across different phages and environmental conditions. Here, we show that the amplification rate of lytic phages can be quantified by analyzing bacterial population growth and collapse dynamics under phage predation using a parsimonious mathematical model — an approach termed Phage-Host Observation for Rate estimation from Collapse Events (PHORCE). We found that the resulting phage amplification rate captures the bactericidal effect independent of initial phage and bacterial population sizes for fast-growing hosts and adsorption-limited phages. Using high-throughput PHORCE, we found that the amplification rates of *Escherichia coli* phages vary widely by more than three orders of magnitude. Furthermore, our approach suggests that phage–antibiotic interactions are predominantly determined by the antibiotic, and not by the phage. In particular, the ribosome-inhibiting antibiotic doxycycline generally showed antagonism with phage amplification, whereas the DNA-damaging antibiotic nitrofurantoin was synergistic. This framework provides a means to quantitatively characterize phage phenotypes and may facilitate future high-throughput phage screens for antibacterial applications.

## Introduction

Advances in metagenomic sequencing have revealed a staggering genomic diversity of phages [1]. A major challenge now is to understand how these diverse phages affect microbial communities, particularly through their ability to kill their bacterial hosts. There are standardized measures of antibiotic activity, such as the minimal inhibitory concentration (MIC) [2]. However, as the phage concentration increases with each predated bacterium, a similar metric for phage antibacterial activity is lacking. This is a pressing problem for potential applications

**Data availability statement:** All relevant data are within the paper and its Supporting Information files. The analysis script can be found at https://doi.org/10.5281/zenodo.14801073.

**Funding:** This work was supported in part by the German Research Foundation (DFG) standalone grant BO 3502/2-1 (to TB) and Collaborative Research Center (SFB) 1310 (to TB). YM was supported by a Marie Curie Fellowship from the European Union and a Humboldt Research Fellowship from the Alexander von Humboldt Foundation. The funders had no role in study design, data collection and analysis, decision to publish, or preparation of the manuscript.

**Competing interests:** The authors have declared that no competing interests exist.

**Abbreviations :** 2D, two-dimensional; BASEL, BActeriophage SElection for your Laboratory; DAOA, double agar overlay assay; LB, Lysogeny Broth; MIC, minimal inhibitory concentration; MOI, multiplicity of infection; OD, optical density; PAS, phage-antibiotic synergy; PFU, Plaque-forming unit; PHORCE, Phage-Host Observation for Rate estimation from Collapse Events.

of phages in therapeutics, the success of which depends on the bactericidal effect of phages [3–5]. This problem extends to the assessment of phage–antibiotic combinations. Indeed, a recent retrospective study showed that phage–antibiotic synergy (PAS) appears to be critical to the success of phage therapies [6]. The identification of synergistic combinations has particular potential, but due to the lack of quantitative measures of phage activity, no consensus quantitative definition of phage–antibiotic synergy has been established. This situation limits a rigorous comparison between treatment options. Clearly, new approaches to quantify phage activity, ideally in high throughput, are urgently needed.

While bacterial population size can be monitored dynamically during growth via optical density (OD), phages must be isolated from their hosts at each time point before quantification [7–9] to determine their kinetic properties [10]. Genomically encoded fluorescent markers could solve this problem, but remain challenging to engineer non-invasively in phages and are therefore rarely used. A promising approach to circumvent this problem is to use growth curves in phage predation assays to infer phage amplification parameters (latent period, adsorption rate, and burst size) through mechanistic mathematical models [9,11,12]. However, these approaches rely on time-resolved measurements of phage concentrations [9,11,12], which severely limits throughput. In contrast, a number of empirical quantifications of phage activity depend only on bacterial growth curves [13–17]. While phage predation experiments are relatively straightforward, commonly used ad hoc measures (e.g., the area under the growth curve [17]) are sensitively dependent on the initial phage and bacterial concentrations. Consequently, cumbersome and error-prone normalization steps are required to enable meaningful comparisons between experiments [18]. Such ad hoc measures are particularly difficult to interpret when comparing phage activity at different bacterial growth rates [18]. As a result, rigorous quantification of phage activity at inhibitory, and therefore clinically relevant, antibiotic concentrations remains a challenge [19].

Here, we present PHORCE, a new approach to determine the bacterial concentration-dependent phage amplification rate directly from bacterial growth curves. This approach allows us to study the phenotypic diversity of lytic phages and to detect phage–antibiotic interactions. We show that the phage amplification rate is a phage-intrinsic kinetic property in that it is insensitive to the initial phage and bacterial concentrations. Importantly, the approach does not require time-resolved phage concentration measurements, opening the door to high-throughput applications. We illustrate the power of PHORCE by measuring amplification rates for the entire BActeriophage SElection for your Laboratory (BASEL) collection of *Escherichia coli* phages [20], revealing that they differ by more than three orders of magnitude and correlate with phage taxonomy. Finally, we show how PHORCE allows for a rigorous definition of phage–antibiotic interactions based on the antibiotic dependence of the phage amplification rate.

## Results

### PHORCE captures phage–host dynamics

We aimed to quantitatively understand the bactericidal activity of phages in a growing bacterial population. To use an assay that could be readily scaled up, we infected an exponentially growing culture of *E. coli* bacteria (K-12 BW25113) with a lytic phage (Bas04, a highly active lytic phage from a publicly available phage library [20]). The bacteria carry a plasmid encoding luciferase genes for accurate growth rate determination (Methods). We measured bioluminescence over time as a proxy for bacterial concentration (Fig 1a). We tracked population size using bioluminescence rather than OD because both are proportional to biomass at steady state, but bioluminescence provides a superior signal-to-noise ratio [21]. Indeed, both

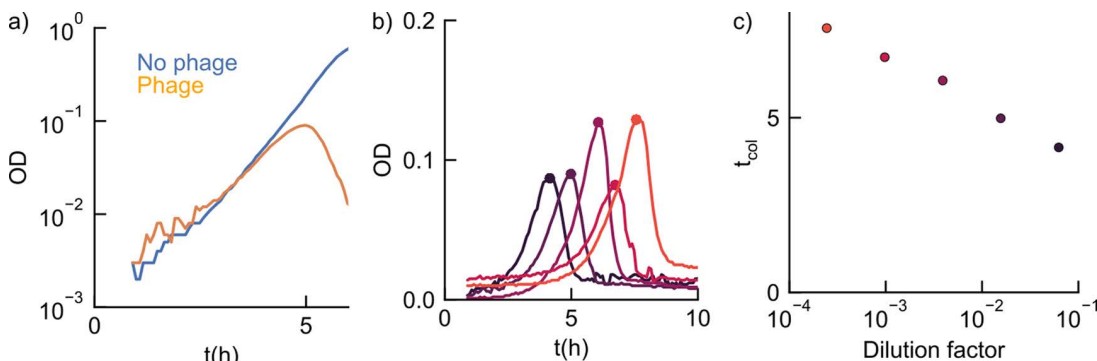

**Fig 1. Phage-induced bacterial population collapse is limited by phage–bacteria adsorption.** (a) Exemplary growth curve of *E. coli* in the absence and presence of a phage (Bas04), measured by bioluminescence (Methods). (b) Growth curves measured at different initial bacterial concentrations $b_0$ ($3 \times 10^3 - 3 \times 10^6$ mL$^{-1}$ from red to black; colors as in **c** at a fixed multiplicity of infection [MOI, i.e., phage-to-bacteria ratio]) of 0.4. (c) Collapse times from **b**; the observed logarithmic decrease (Pearson's $\rho = 0.995$, $p = 10^{-10}$) suggests that the phage amplification kinetics is adsorption-limited (Methods). Error bars show standard errors from bootstrapping the growth curves. We obtained similar results with bioluminescence as with optical density (S1 Fig ). The data underlying this figure can be found in S1 Data.

read-outs report similar dynamics (Pearson's $\rho = 0.99$, $p = 10^{-65}$, S1 Fig). We observed that the initial growth of bacteria with and without phages is similar, in particular exhibiting the same growth rate (Fig 1a). This is in contrast to most antibiotics, which rapidly decrease the bacterial growth rate at sufficient concentrations. However, after 8 h, the bacterial population suddenly collapsed in the presence of phage (orange), while the no-phage control (blue) maintained its exponential growth. This sudden collapse, which has been observed in many previous studies using lytic phages [13–15], is due to phage predation and therefore depends on the bactericidal effect of a phage: At the time of collapse, the phage population is large enough to lyse bacteria at a higher rate than they can grow.

Successful phage predation consists of two critical steps: First, a phage adsorbs to the bacterial host. After a latency period required for phage replication within the host, the second step occurs: bursting of the infected bacterium and release of progeny phage particles. To model this process, we first investigated which of these two steps (adsorption or latency) is rate-limiting for the infection cycle under our conditions. Importantly, the adsorption rate depends on the bacterial concentration, whereas the latent period typically does not. We reasoned that this distinction could be exploited to identify the rate-limiting step in our assay: If only latency is limiting, then the collapse time should be independent of the absolute concentrations of bacteria and phages and only depend on their ratio (Methods). To test this directly, we used the growth curve assay to measure collapse times for a wide range of initial bacterial densities (Fig 1b) while maintaining a fixed bacterium-to-phage ratio (multiplicity of infection [MOI]). Instead of a constant collapse time, we observed a precise logarithmic decrease in collapse time with initial bacterial concentration over three orders of magnitude (Fig 1b, 1c). These results indicate that, under our conditions, adsorption is rate-limiting for the phage infection cycle (Methods).

Trying to quantitatively explain our observations, we analyzed a minimal mathematical model of phage predation [12] (Fig 2a). This model consists of coupled differential equations describing the bacterial density $b(t)$ growing at a rate $r_{\mathrm{bac}}$ and the phage density $p(t)$ adsorbing to bacteria at a rate $k_{\mathrm{adsorb}}$. With each phage binding cycle, $r_{\mathrm{burst}}$ phages are created and one bacterium disappears (Methods):

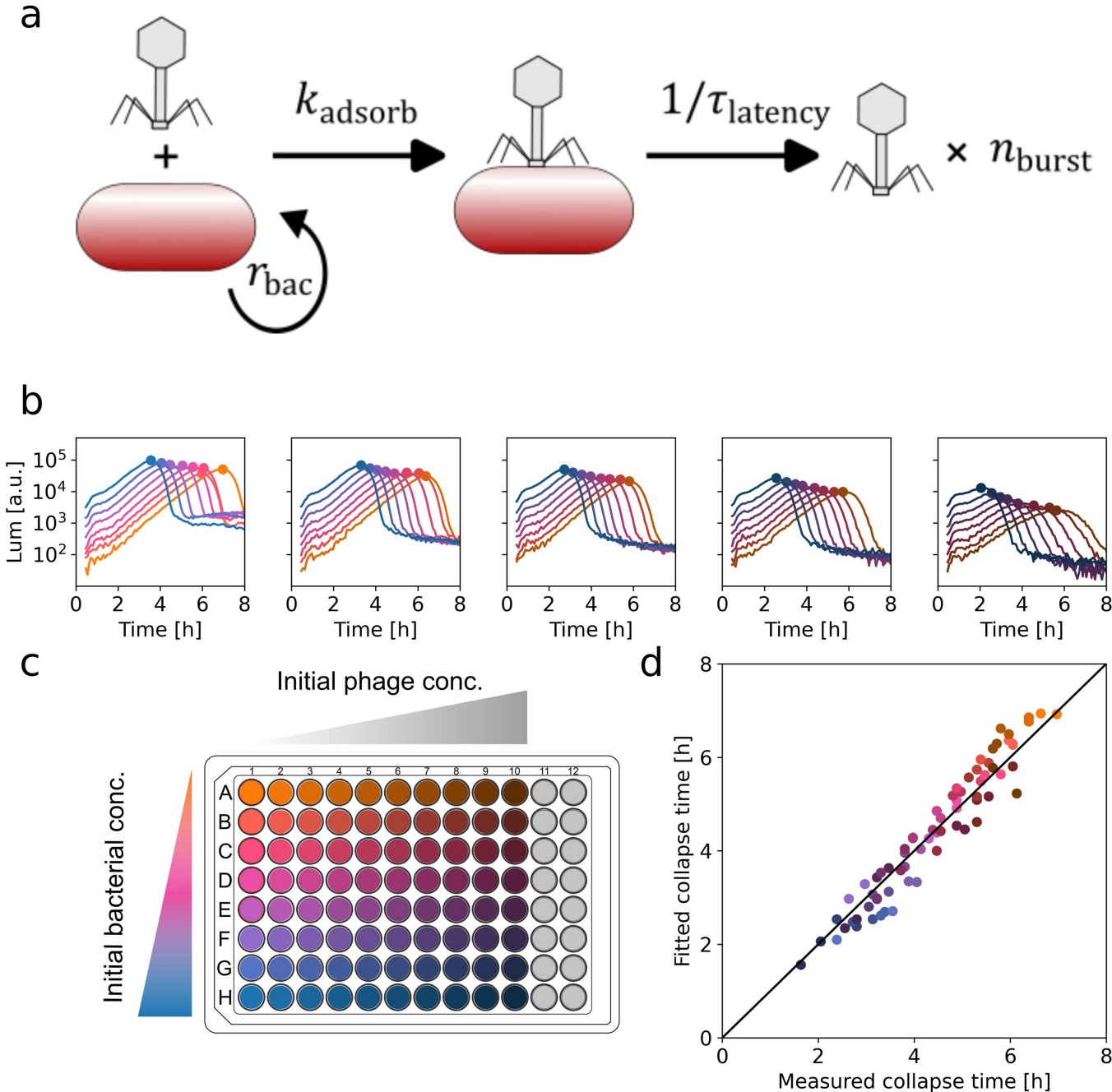

**Fig 2. A mathematical model of phage infection allows the extraction of key phage amplification parameters (PHORCE). (a)** Schematic of minimal model of phage amplification (Methods). We approximate the process by a single parameter: the phage amplification rate $r_{pha} = n_{burst}k_{adsorb}$. **(b–c)** *E. coli* growth curves measured by bioluminescence (Methods) in the presence of the phage Bas04 at all combinations of eight different initial bacterial concentrations (1.1 × 10⁴–1.4 × 10⁶ mL⁻¹, orange to blue) and 10 different initial phage concentrations (2.5 × 10³–1.3 × 10⁶ mL⁻¹, light to dark). Due to the large number of growth curves, only every other phage concentration is shown. Circles indicate the collapse points. **(d)** Comparison of observed collapse times and collapse times from the model. Using Eq 2, we fit all collapse times measured in panel **a** simultaneously with a single value of $r_{pha}$ as the only free parameter; the model is consistent with the experimental data ($r^2 = 0.93$). The data underlying this figure can be found in S1 Data.

$$\frac{\mathrm{d}b}{\mathrm{d}t} = r_{\mathrm{bac}} b(t) - k_{\mathrm{adsorb}} p(t) b(t) \tag{1a}$$

$$\frac{\mathrm{d}p}{\mathrm{d}t} = n_{\mathrm{burst}} k_{\mathrm{adsorb}} p(t) b(t) \tag{1b}$$

In this regime of adsorption-limited phage predation, we derive an analytical result for the collapse time $t_{\mathrm{col}}$ (Methods):

$$t_{\mathrm{col}} = \frac{1}{r_{\mathrm{bac}}} \log\left(1 + \frac{r_{\mathrm{bac}}}{r_{\mathrm{pha}} b_0} \log\left(\frac{p_\infty}{p_0}\right)\right) \tag{2}$$

Here, $r_{\mathrm{pha}} = n_{\mathrm{burst}} k_{\mathrm{adsorb}}$ is the product of the adsorption constant $k_{\mathrm{adsorb}}$ and the burst size $n_{\mathrm{burst}}$ (Methods), which plays the role of an amplification rate for phages in the adsorption-limited regime (see below). $p_0$ and $p_\infty$ denote the initial and final phage concentrations, respectively, $b_0$ is the initial bacterial concentration, and $r_{\mathrm{bac}}$ is the bacterial growth rate. Note that mathematically, the product of $r_{\mathrm{pha}}$ and the bacterial concentration gives a specific population growth rate, analogous to the well-established bacterial growth rate. In essence, $r_{\mathrm{pha}}$ quantifies how fast the phage population can grow in a bacterial population and conveniently characterizes the double-exponential amplification of phages in an exponentially growing bacterial population, regardless of the bacterial concentration. Therefore, we refer to $r_{\mathrm{pha}}$ simply as the *phage amplification rate*. Unlike parameters previously used to characterize phage amplification, the phage amplification rate defined in this way is independent of bacterial and phage concentrations [13–15,18].

Our result for the collapse time (Eq 1) provides an efficient way to test whether the mathematical model is consistent with the dynamics observed in phage predation experiments. To this end, we next performed experiments in which we systematically varied initial bacterial densities over two orders of magnitude (Fig 2b, 2c, colors) and initial phage densities over three orders of magnitude (Fig 2b, 2c, columns) and quantified the collapse times for the two-dimensional (2D) array of all 80 pairwise combinations. Next, we estimated the fold increase in phage population size $p_\infty / p_0$ from subsequent measurements of bacterial growth curves ([9], Methods, S2 Fig). This rich, quantitative dataset enables a powerful test of the model's prediction of collapse time since the phage amplification rate remains as the sole free parameter in Eq 1.

We found a quantitative agreement between the model and the experimental data over the entire range of initial conditions for a single value of the phage amplification rate (Fig 2d, $\rho$ = 0.9). These results support that the mathematical model faithfully captures the essence of the phage predation dynamics over several orders of magnitude of initial phage and bacterial densities – up to the point of collapse (S3 Fig). Post-collapse dynamics are largely dominated by resistance evolution (S4 Fig) and beyond the scope of this model. While we have only tested lytic phage activity, simulations suggest that the model works similarly well for lysogenic phages, as lysogeny predominantly affects the post-collapse dynamics (S4 Fig and [9]). Furthermore, we found that the model works not only for *E. coli* laboratory strains (Fig 2d) but also for a clinical isolate of *Pseudomonas aeruginosa* infected with a therapeutically used phage (S5 Fig). While the latent period may also contribute to the dynamics, assuming adsorption as the sole rate-limiting step already quantitatively explains our results. This approach also illustrates that the phage amplification rate can be readily measured without the need for considerably more laborious time-resolved measurements of phage concentration [9,11,12]. We have

termed this approach to quantify phage amplification rates Phage-Host Observation for Rate estimation from Collapse Events (PHORCE).

## Phage amplification rates show great diversity

To test the broader applicability of PHORCE, we applied it to the entire BASEL collection of 69 different lytic *E. coli* phages [20] (Fig 3a). These phages are ideally suited for this purpose since they all infect the *E. coli* strain used in this study, are thought to represent natural *E. coli* phage diversity, and are well characterized phenotypically and genetically [20]. Bacterial growth curves for similar initial phage and bacterial densities showed that the collapse time is strongly dependent on the phage used, as expected (Figs 3b and S6). We confirmed the main observations made above with the Bas04 phage (Figs 1 and 2): The collapse time generally decreases with the initial bacterial concentration at a fixed MOI (S7a Fig), and the minimal

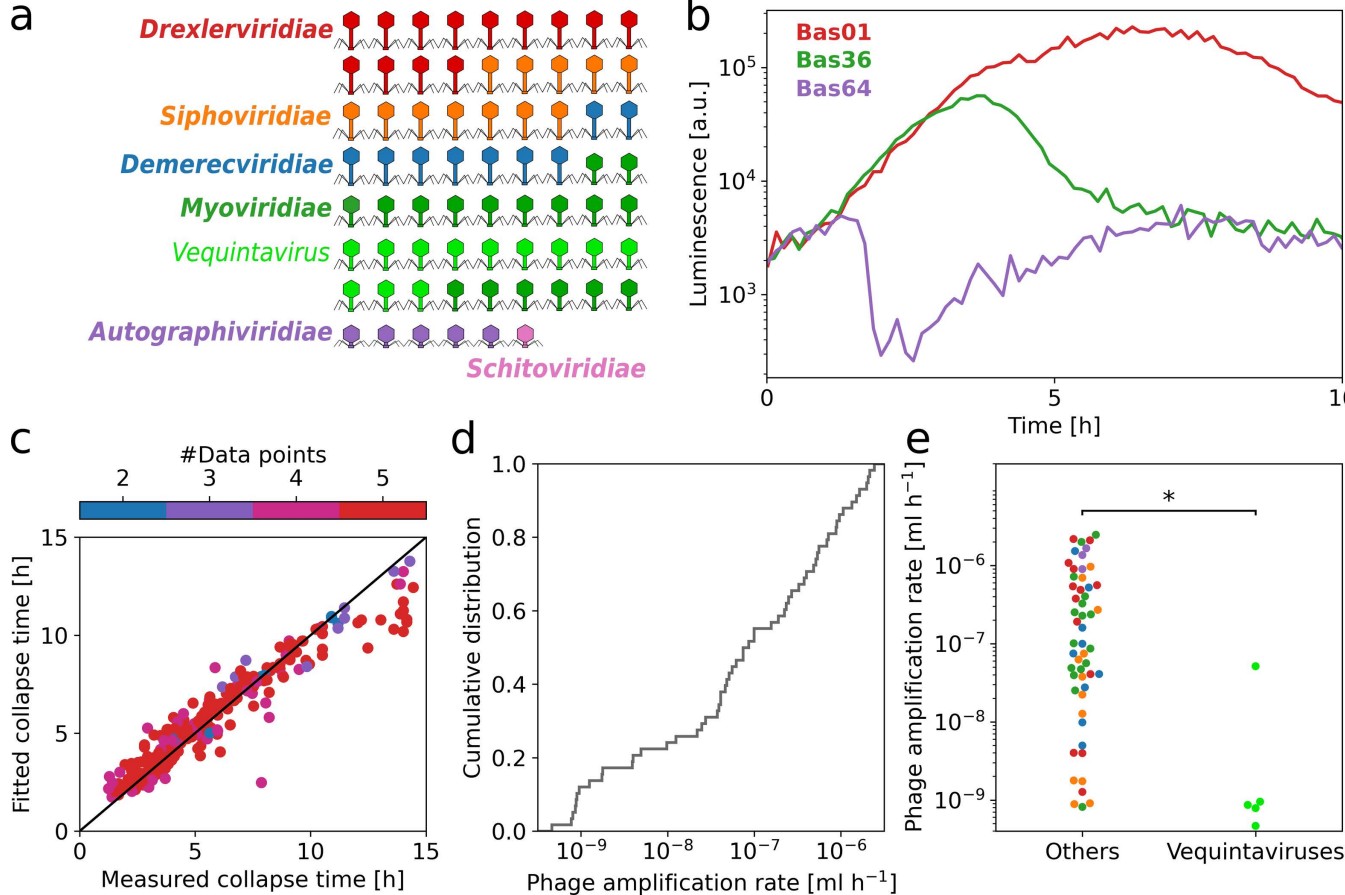

**Fig 3. High-throughput PHORCE reveals an extremely wide range of phage amplification rates for a collection of diverse lytic phages. (a)** Schematic of 69 phages in the BASEL collection showing the phage morphology and taxonomic classification (colors). **(b)** Example growth curves in the presence of different phages (colors) at the same initial phage ($1 \times 10^5$ mL$^{-1}$) and bacterial concentration ($4 \times 10^5$ mL$^{-1}$) with different collapse times. **(c)** The fitted and measured collapse times for all BASEL phages, similar to Fig 2d, show good agreement (Pearson's $\rho = 0.95$, $p = 10^{-129}$), regardless of the number of fitted data points (color). **(d)** Distribution of the phage amplification rates for the entire BASEL collection (Methods). **(e)** Vequintavirus members have a significantly lower amplification rate ($p = 8 \times 10^{-5}$, Mann–Whitney *U* test with Bonferroni correction for multiple testing) than other phages. The detection limit for the phage amplification rate was $3 \times 10^{-10}$ mL h$^{-1}$, as slower-growing phages did not cause a collapse within the exponential growth regime of the bacteria. This was the case for 7 of 12 *Vequintavirus* members and 4 of 57 others; these phages were excluded from the analysis. The data underlying this figure can be found in S1 Data.

model of phage predation in the adsorption-limited regime faithfully captures the dynamics for the entire BASEL collection using the phage amplification rate as the sole fit parameter (Fig 3c). These results demonstrate that PHORCE is applicable to diverse lytic phages and can be readily used as a method for quantifying phage amplification rates in high-throughput assays.

Applying PHORCE to all phages in the BASEL collection revealed an enormous diversity of phage amplification rates. Rates varied by more than three orders of magnitude among the phages (Fig 3d), and the true diversity is even greater because some phage amplification rates were below the detection limit of our technique. To put this diversity into perspective, the amplification rates of the host of these phages (i.e., different *E. coli* strains) vary by less than an order of magnitude in common growth media, and even the growth rate ratio between the fastest-growing known bacterium, *Vibrio natriegens (*10 min doubling time [22]), and the notoriously slow-growing *Mycobacterium tuberculosis (*24 h doubling time [23]) is more than 10 times smaller than the observed range of phage amplification rates. We expect the diversity of phage amplification rates to increase even further when other bacterial hosts are considered.

We found that the phage genus partially determines the phage amplification rate. In particular, *Vequintavirus* members amplify significantly slower than other phages ($p = 3 \times 10^{-4}$, Fig 3e). However, we did not find any relationship between amplification rate and other phage characteristics, such as genome size or primary/terminal receptor (S1 Table). Of particular interest is phage morphology, which has traditionally been used to classify phages. In contrast to the phage genus, phage morphology had no predictive value for phage amplification rate (S7b Fig). This lack of association between morphology and phenotype further supports the plausibility of the modern phage classification based on sequence rather than morphology.

How does the phage amplification rate compare to other measures of phage efficacy? A common measure of phage efficacy is the total number of phages present in the culture at the end of a phage predation experiment, e.g., determined by a plaque assay [24,25]. While this phage yield is clearly distinct from the phage amplification rate, it is possible that in practice these two quantities are highly correlated and measure essentially the same underlying property. We have already shown that the phage amplification rate faithfully captures the collapse time of bacterial populations across a wide range of initial conditions (Figs 2d, 3c), but phage yield did not correlate at all with this key quantity (S7c Fig). These results suggest that in addition to conceptual advantages such as independence of initial phage and bacterial densities, phage amplification rate can be a useful measure to quantify phage efficacy for potential therapeutic applications aimed at eradicating bacterial populations.

## Phage–antibiotic interactions are largely determined by antibiotics

Since PHORCE can disentangle phage amplification rates from bacterial growth rates, we reasoned that it could be suitable for characterizing interactions between phages and antibiotics. Antibiotics can interfere with the ability of phages to infect bacteria; a case in point is aminoglycosides which have recently been shown to be potent inhibitors of phage infection [25]. Our approach should readily detect such phage–antibiotic interactions, which are analogous to drug interactions [21]. A key difference is that drug interactions between antibiotics are typically defined in terms of the drugs' quantitative effects on the bacterial growth rate [21]. This approach is not directly applicable to phage–antibiotic interactions, because phages do not simply lead to exponential growth with a lower rate, but rather to more complicated population dynamics (Fig 1). In a plausible definition of phage–antibiotic interactions, antagonism leads to a decrease in phage amplification rate in the presence of the antibiotic and synergy to an increase; in a neutral interaction, the antibiotic does not affect the phage amplification rate.

The fact that the phage amplification rate is independent of initial conditions and easy to measure is valuable for the rigorous definition and quantification of phage–antibiotic interactions.

To test this idea, we measured phage amplification rates in the presence of antibiotics at different concentrations. We first focused on the phage Bas04 and a fine-resolution concentration gradient of the ribosome-inhibiting antibiotic doxycycline. We set up several parallel precultures at different antibiotic concentrations to ensure steady-state growth, before adding phages. The phage-induced collapse time showed a clear monotonic increase with doxycycline concentration, while the bacterial population size (measured by luminescence intensity, Methods) at which the collapse occurred remained nearly constant upon drug addition (Fig 4a, blue). The phage amplification rate derived from these data (Eq 11) decreased with increasing antibiotic concentration (Fig 4b, blue), revealing a clear antagonistic interaction between Bas04 and doxycycline. This antagonism is not a general consequence of lowering the bacterial growth rate, because the interaction of this phage with the DNA-damaging drug nitrofurantoin is strikingly different: Here, the collapse time remained nearly constant and the bioluminescence intensity at the time of the collapse increased rapidly with drug concentration. The phage amplification rate increased monotonically with the drug concentration (Fig 4a, 4c, orange), revealing a clear case of synergy between Bas04 and nitrofurantoin. These results show that antibiotics can drastically affect phage amplification in a synergistic or antagonistic manner, which can be detected using PHORCE.

To determine whether the observed interactions between antibiotics and phages are specific to each phage or reflect more general effects of the drugs on phage infection, we analyzed phage–antibiotic interactions for a diverse selection of phages. We measured the amplification rate of nine different phages from the BASEL collection in fine-resolution concentration gradients of nitrofurantoin and doxycycline. Similar to Bas04 above, these experiments showed clear effects of the antibiotics on the dynamics of bacterial population collapse for most phages (S8 Fig). To systematically detect phage–antibiotic interactions, we analyzed the correlation between phage amplification rate and drug concentration for each phage–antibiotic combination (Fig 4b). Since we do not know the functional form (e.g., linearity) of the antibiotic-dependent phage amplification rate, we used Kendall's $\tau$ as the interaction score, with $\tau > 0$ indicating synergy, and $\tau < 0$ antagonism, and $p < 0.05$ as the significance threshold. The interaction score strictly identifies only the direction (synergy or antagonism), but we found that it correlates with the strength of the interaction (Kendall's $\tau$ = 0.5, $p$ = 0.002, S9a Fig).

For doxycycline, this analysis revealed three cases of neutrality ($p < 0.05$) and no case of synergy; the majority of interactions (six out of nine) were antagonistic (Fig 4d). In contrast, the majority of phages (six out of nine) showed synergy with nitrofurantoin while not a single case of antagonism was detected for this drug (Fig 4d). Thus, phages that showed any interaction at all generally synergized with nitrofurantoin but were antagonized by doxycycline. Additionally, the interaction scores for both drugs were not correlated across different phages (Kendall's $\tau$ = 0.1, $p$ = 0.6; Fig 4e). Taken together, these results indicate that phage–antibiotic interactions are primarily determined by the antibiotic and less by the phage.

Previous studies of phage–antibiotic interactions have mostly focused on synergy and often relied on endpoint measurements of phage yield [24,25]. We tested the extent to which these methods correlate with our approach, which is based on the dynamics of bacterial population growth and collapse. While there is a clear difference in the bacterial growth curves between nitrofurantoin and doxycycline for Bas04 (Fig 4a), the phage yield decreases similarly with increasing concentration for both antibiotics (Fig 4c). In fact, we found no correlation between the antibiotic dependence of phage yield and the antibiotic dependence of phage amplification rate across all phages and drug concentrations for both antibiotics (Kendall's

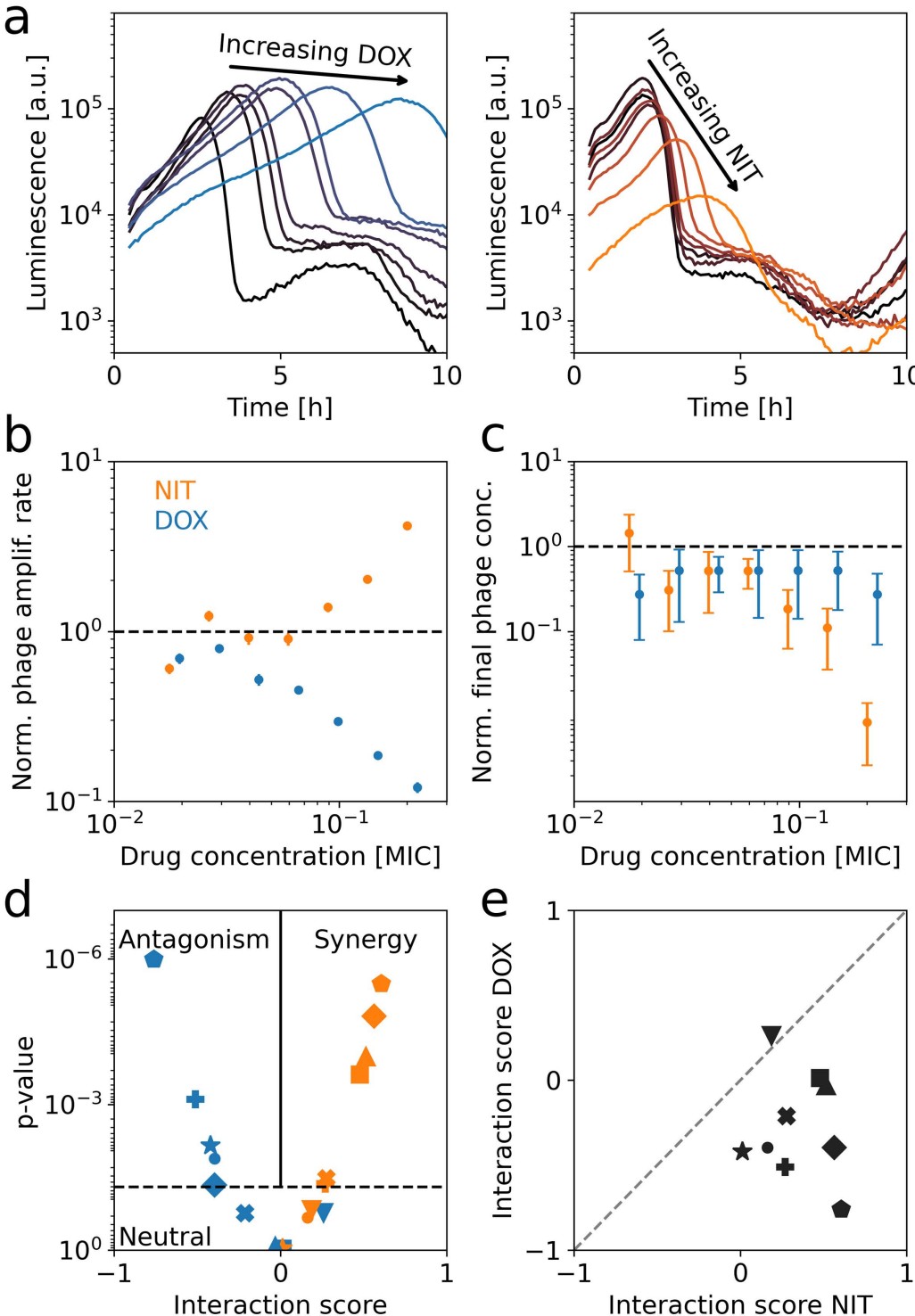

**Fig 4. Phage–antibiotic interactions are determined by the antibiotic, not the phage.** (**a**) Growth curves of bacteria and phages (Bas04) in the presence of doxycycline (left) and nitrofurantoin (right). (**b**) Phage amplification rate of Bas04 at different concentrations of nitrofurantoin (orange) and doxycycline (blue). While the phage amplification rate increases with nitrofurantoin (synergy), it decreases with doxycycline (antagonism). Error bars show propagated standard errors from bootstrapping the growth curves (typically smaller than the marker size). (**c**) As **b**, but for phage yield (Methods). In contrast to the phage amplification rate, the yield is not markedly different between the two drugs. Error bars show propagated standard errors from bootstrapping the growth curves. (**d**) Interaction score of nine

different phages with both nitrofurantoin (orange) and doxycycline (blue); interaction score is defined as Kendall's $\tau$ of the antibiotic-dependent phage amplification rate (as in **c**). The dashed line is at $p = 0.05$; positive correlations indicate synergy, and negative values indicate antagonism. Nitrofurantoin shows significantly more synergistic interactions than doxycycline ($p = 0.008$, Wilcoxon signed-rank test). (**e**) Scatterplot comparing interaction scores of different phages with nitrofurantoin to those with doxycycline. No correlation was found between the two drugs ($p = 0.9$, Kendall's $\tau$), showing that the antibiotic, but not the phage, is predictive of the interaction. Dashed line indicates identity. Phages in **d,e**: Bas01 (■); Bas04 (●); Bas10 (+); Bas14 (▼); Bas23 (×); Bas30 (★); Bas37 (⬟); Bas42 (◆); Bas63 (▲). The data underlying this figure can be found in S1 Data.

$\tau = 0.05$, $p = 0.9$, S9b Fig). Together with the lack of correlation between phage amplification rate and phage yield across the entire BASEL collection (S7c Fig), these results show that the phage amplification rate is a distinct property of phages that is largely independent of the commonly used phage yield. While endpoint measurements of phage yield are certainly relevant in other contexts, the phage amplification rate quantitatively captures the bactericidal effect of phages across diverse conditions (Figs 2d, 3c) and is thus a key quantity relevant to PAS.

## Discussion

We have shown that phage amplification under our conditions is limited by phage–bacteria adsorption (Fig 1), enabling characterization of phage bactericidal effect by a single parameter, the phage amplification rate. This rate is readily measured with PHORCE and is insensitive to the initial phage and bacterial concentrations (Fig 2). Applying PHORCE to the BASEL collection of diverse *E. coli* phages revealed that the phage amplification rate varies by more than three orders of magnitude (Fig 3). Using this parameter to quantitatively characterize phage–antibiotic interactions, we observed that doxycycline antagonized many phages, while nitrofurantoin often synergized with them (Fig 4).

Similar to the specific growth rate of bacteria, which is one of the cornerstones of microbiology, phage amplification rate provides a key phage characteristic that can be directly compared across different phages, hosts, and growth conditions. This is difficult to achieve with empirical quantification [13–15,18]. While plaque size measurements have been used to detect bactericidal effect [26], they are hard to standardize across experiments and laboratories due to their sensitivity to initial conditions [27]. In addition, PHORCE does not require time-resolved measurements of phage concentrations, resulting in greatly increased throughput compared with previous mechanistic models [9,11,12]. Previous studies on PAS have often relied on the phage yield [24,25], which does not correlate with bactericidal effect under our conditions (S7c and S9b Figs). Instead, we propose to use the phage amplification rate as the primary measure of a phage's bactericidal effect, similar to how the MIC is used to assess the bactericidal effect of antibiotics.

The phage amplification rate depends on the culture conditions. For example, we see a strong dependence on antibiotic concentration (Fig 4), and we speculate that there are differences between, e.g., rich and minimal media or at different temperatures. This is similar to the situation in antibiotic susceptibility testing, where the MIC is affected by growth conditions [2]. Using preclinical infection models, it will be interesting to see which culture conditions best predict *in vivo* activity as a first step toward mimicking patient conditions. In addition, it will be interesting to perform cross-laboratory validation of reproducibility under similar culture conditions.

Notably, we found no correlation between the phage amplification rate and the phage yield (S7c Fig). While one might intuitively expect that a higher phage amplification rate would

lead to a higher phage yield, it should be noted that a faster collapse of the bacterial population caused by a higher phage amplification rate also implies a lower bacterial population at the time of collapse and thus a lower phage yield. For future work, it will be interesting to model and experimentally verify the dependence of the phage yield on phage kinetics. With such a model in hand, we can also begin to understand any antibiotic-dependent phage yield.

PHORCE relies on the assumption of adsorption-limited phage dynamics, which we verified experimentally (Fig 1c). This assumption holds for all *E. coli* phages we tested (S7 Fig) and also for a *P. aeruginosa* phage (S5 Fig). However, this does not necessarily apply to all growth conditions or all bacteria. For example, for slow-growing bacteria such as pathogenic mycobacteria, the binding rate does not necessarily decrease, but the slower bacterial metabolism is likely to increase the latent period. Therefore, it will be interesting in future work to test the validity of our model for slow-growing bacteria.

PHORCE also relies on a detectable collapse of growth curves in liquid medium. However, some phages show plaques on bacterial lawns but do not noticeably affect liquid growth curves. In the *E. coli* phage collection we tested, this was the case for 11 out of 69 phages. Phages showing plaques on solid agar but no collapse in liquid media have previously been attributed to potential different infection states occurring in liquid [28], and it would be interesting to experimentally verify the existence of such states. We propose that an alternative explanation for this discrepancy between agar and liquid media is that these phages have such a low amplification rate that the critical phage density to cause a collapse is not reached before the bacterial culture reaches stationary phase. For such phages, it will be interesting to develop a similar model for extracting phage kinetic parameters from experimental data, using time-resolved plaque growth data instead of growth curves in liquid media.

In a mathematical model of phage therapy, the key parameter predicting bacterial clearance is the ratio of the phage clearance rate through the body to the product of the phage burst size and the adsorption rate [29]. Our work shows how this product of burst size and adsorption rate, which we defined as the phage amplification rate, can be readily measured in simple laboratory experiments. The rate at which phages are cleared from the body cannot be measured in vitro; future work should focus on measuring these rates *in vivo* to select phages with the highest potential for achieving bacterial clearance.

While our model accurately captures the dynamics up to the point of collapse, it does not capture the rate of the collapse nor does it capture the regrowth of resistant mutants (S3 Fig). Recently, it has been shown that the collapse rate is determined by the latent period [9], suggesting that this parameter can also be extracted from bacterial growth curves. Finally, resistance evolution is a major problem in phage therapy [30]. Therefore, an important future direction is to quantitatively extract the rate of resistance evolution from the post-collapse growth curves, so that the optimal phages can be quantitatively evaluated for both their lytic activity and how rapidly they induce resistance.

The enormous diversity in phage amplification rates we observed (Fig 3) has a number of implications. Phages play a vital role as bacterial predators in various ecosystems such as the ocean, gut, and soil, for which recent studies have extensively characterized phage–host networks of infectivity [31–33]. Our findings highlight large variation in bactericidal effect among phages. This implies that phage–host interactions occupy a continuum of activity, rather than the simple binary yes–no classification often used [31–33]. This continuum of phage activity also has potential implications for phage therapy. While some personalized phage therapies consider quantitative measures of bactericidal effect [34], off-the-shelf cocktails typically rely on binary measures of phage–host interactions [35–37], in part due to a lack of easy-to-use techniques for comparing phage activity across hosts. This is a missed opportunity, as highlighted by a strong correlation between phage amplification rate and therapeutic

efficacy recently observed in a *Drosophila melanogaster* infection model [18]. Measuring phage amplification rates has traditionally required laborious time-resolved measurements of phage concentration [5]. Our high-throughput approach eliminates the need for such time-consuming measurements, using standard equipment that can be seamlessly integrated into product development pipelines.

Phage cocktail design can be further aided by detecting phage–phage interactions. The model can provide a quantitative null hypothesis of what a bacterial growth curve should look like in the presence of two different phages, based on the individual phage–bacteria growth curves, assuming no-phage–phage interactions. Any deviations from the model predictions would indicate interactions. In future work, we plan to extend the model for multi-phage predation and perform experiments to detect phage–phage interactions, which can then be analyzed using, e.g., RNA-seq.

Using PHORCE, we observed a notable pattern: For the phage–antibiotic pairs we tested, phage–antibiotic interactions are more strongly determined by the antibiotic rather than by the phage itself (Fig 4d and 4e). This suggests a mechanism whereby antibiotics have a broad effect on the ability of phages to replicate inside bacterial cells. For example, doxycycline inhibits ribosomes [21], which likely halts phage production because phages cannot hijack them as usual [38]. This could explain the widespread antagonism observed with doxycycline (Fig 4). Nitrofurantoin's mechanism of action is more intricate and involves the reduction of this prodrug to reactive intermediates that damage various cellular components; ultimately, resulting in decreased transcription [39]. We speculate that this reduced transcription and subsequent reduced load on the protein synthesis machinery may facilitate phage replication by hijacking this machinery. Investigating the underlying mechanisms of phage–antibiotic interactions is an interesting avenue for future work. As a starting point, it would be valuable to explore whether phage–antibiotic interactions are generally determined by the drug's mode of action, by extending the approach presented here to measure a comprehensive phage–antibiotic interaction network for a wider range of phages and a diverse set of antibiotics.

Our approach has applications beyond phage–antibiotic interactions and phage cocktail design. Phages are typically classified as generalists or specialists based on their ability to infect a wide or narrow range of hosts, respectively [40]. A fundamental question is whether generalism is associated with a fitness cost. Measurements of phage amplification rate are well suited to directly address such fitness tradeoffs. Since amplification rate is a key aspect of phage fitness, we see another application of PHORCE in evolutionary biology. Phages, with their large numbers, short generation times, and compact genomes are a popular model system in laboratory evolution [41]. While their evolution is nowadays relatively easy to characterize genotypically, there is a lack of quantitative methods to assess their phenotypic evolution. PHORCE can fill this gap. Finally, the unprecedented ease with which phage amplification rates can be measured (S1 Text) opens up the possibility of systematically investigating quantitative relationships between these amplification rates and various aspects of phage immunity [42] and bacterial physiology [43].

## Materials and Methods

### Bacterial strains and bacteriophages

All collapse curve experiments were performed with *E. coli* K-12 BW25113 transformed with a kanamycin-resistance-bearing plasmid (pCS-λ) carrying luciferase genes used to determine the growth rate [21]. Previously, we have shown that this plasmid is stably maintained in the bacteria for many generations, even in the absence of the selective antibiotic [21]. The 2D gradient experiments (Figs 1, 2, S1 and S2) employed the Bas04 bacteriophage, sourced from

the BASEL collection [20]. For the antibiotic-phage interactions (Figs 4, S8 and S9), a subset of nine phages (Bas01, Bas04, Bas10, Bas14, Bas23, Bas30, Bas37, Bas42, and Bas63) was chosen to represent a wide diversity of phage amplification rates and phage families. Additionally, *E. coli* K-12 MG1655 ΔRM, an isolation strain from the BASEL collection, was used for conducting plaque assays to determine phage titers and for the generation of phage stock solutions. Purified phage stocks were stored at 4 °C, whereas bacterial stocks were stored at −70 °C with the addition of glycerol (17% in final solution). Both were stored in CryoPure tubes (Sarstedt, 72.377.992).

## Growth media

For all collapse curve and reinoculation experiments, Lysogeny Broth (LB) Lennox medium (Sigma Aldrich, L3022) enriched with 5 mM $MgSO_4$ (Sigma Aldrich, M7506) was utilized. The plaque assays employed the double agar overlay assay (DAOA) technique [7], which required an underlay of 1.5% LB agar (Sigma Aldrich, L2897) and an overlay of 0.5% LB agar, both supplemented with 5 mM $MgSO_4$. In the DAOA process, bacterial overnight culture was added to the overlay to an OD of 0.4 within the soft agar.

The antibiotics used were doxycycline (Sigma Aldrich, D9891) and nitrofurantoin (Sigma Aldrich, N7878). The doxycycline stock solution was prepared by dissolving it in DMSO (Sigma Aldrich, D8418), while nitrofurantoin was dissolved in DMF (Sigma Aldrich, D4551). Both solutions were then filter-sterilized and stored at −20 °C in the dark. Aliquots were thawed at room temperature before use.

## Two-dimensional gradient of initial phage and bacterial concentration

We set up phage predation assays with a 2D gradient of initial bacterial and phage concentrations (Fig 2). For the bacterial preculture, a frozen overnight culture stock was diluted 1:500 in fresh medium and incubated for 2–3 h in a shaking incubator (Innova 44, Eppendorf New Brunswick, 30 °C, 225 rpm). We use frozen stocks of overnight cultures for simplicity and because experience in our laboratory shows that this generally gives nearly identical results and usually increases reproducibility. Note that we also ensure that the bacterial cultures are in a steady state of exponential growth by preculturing for several hours. A gradient of phage titers was prepared in the same medium using dilutions by a factor of two. The bacterial concentration gradient was established orthogonally, also using dilutions by a factor of two of the preculture, and added on top of the phage gradient in a white microtiter plate (LumiNunc F96-MicroWell, Thermo Fisher Scientific, 732−2697). The resulting final culture volume was 140 μL per well, and phage dilutions ranged from $5 \times 10^{-3}$ to $10^{-7}$ from a phage stock concentration of $2.6 \times 10^8$ pfu/mL in 10 steps and bacterial dilutions ranged from $10^{-2}$ to $8 \times 10^{-5}$ in eight steps from a preculture of $1.4 \times 10^8$ bacteria/mL (Fig 2). Plates were sealed with transparent foil (TopSeal-A Plus, PerkinElmer, 6050185) and placed in a plate reader (Synergy H1, BioTek, VT, USA) at 30 °C and 807 rpm, with bioluminescence measurements taken every 5 min. Dual acquisition of both OD and bioluminescence (Figs 1 and S1) was achieved by performing experiments in black plates with a transparent bottom (CELLSTAR Ref. No. 655087); in this experiment, data points were acquired every 20 min. In addition, the cell density of the preculture was determined using a Coulter counter as described below.

## Antibiotic concentration gradients

We set up phage predation assays in concentration gradients of two different antibiotics, doxycycline and nitrofurantoin (Figs 4, S8, and S9). For each antibiotic, an eight-step concentration gradient was constructed using a 1.5-fold serial dilution, with maximum concentrations

between 3 and 4 µg/mL. Precultures were inoculated in these antibiotic concentration gradients by diluting frozen stocks between 1:1,500 and 1:3,000 and incubated for 4 h at 30 °C (225 rpm for doxycycline and 807 rpm for nitrofurantoin). After this incubation, the precultures were re-diluted (5-fold to 10-fold) in prewarmed medium containing the respective antibiotic concentration and transferred to a white microtiter plate. The cell density of the precultures was determined using a Coulter counter (see below). For each phage–antibiotic interaction measurement, one of the 10 tested phages was added to each antibiotic concentration at a $10^{-5}$ dilution. One column of the antibiotic concentration gradient plate served as the phage-free control. The plate was sealed with transparent foil and incubated in a plate reader at 30 °C and 807 rpm, monitoring bioluminescence every 5 min for 17–20 h. Note that the determination of collapse time is insensitive to any potential effect of the antibiotic on the bioluminescence signal per cell.

## Phage purification from bacterial culture

For phage purification, 130 µL of a phage–bacteria mixture was combined with 130 µL of medium and approximately 15 µL of chloroform (Sigma Aldrich, 372978) in each well of a polypropylene 96-well deep well plate as chloroform dissolves polystyrene plates. The plate was then subjected to vigorous shaking at approximately 1,200 rpm using a Titramax 1,000 platform shaker (Heidolph, 544-12200-00) for a duration of 5–10 min. Following this, an additional 1,040 µL of medium was introduced to each well, and the plate was shaken again, this time at a lower speed of 600 rpm, for 2 min to ensure thorough mixing. In the next step, the plate was centrifuged at 2,580 rpm for 30–40 min, aiming to sediment the cell lysate at the bottom of the wells. After centrifugation, a 150 µL aliquot of the supernatant from each well was carefully transferred to a new 96-well microtiter plate. Due to the addition of medium during the purification process, the resulting product corresponds to a 1:10 dilution of the pure phage lysate.

## Measurement of final phage concentrations

We measure final phage concentrations (Figs 4c and S7c) by purifying the product of the phage predation assays (see section "Phage purification"), reinoculating it with bacteria and comparing the collapse times with a dilution series of the respective phage stock. For initiating the bacterial preculture, 2 mL of medium was inoculated with 4 µL of a frozen overnight culture stock in a standard glass test tube. This mixture was then incubated in a shaking incubator at 30 °C and 225 rpm for 2–3 h. Following incubation, the preculture was diluted in a range from 1:60–1:300 with fresh medium and aliquoted to a microtiter plate, dispensing 138.6 µL into each well. Each well then received 1.4 µL of either the phage purification product (at a 1:10 dilution) for reinoculation or an equivalent volume of fresh medium for control wells. To construct calibration curves, one column per phage strain was used, starting with a phage stock dilution of 1:100 in the top well and proceeding with a 1:3 dilution series down the column. The plate was subsequently sealed, and bioluminescence was measured as described above. Because phage yield is measured after an overnight incubation, resistant bacteria sometimes emerge. However, this does not have a major effect on the phage yield (S10 Fig).

## Quantification of cell densities

For Coulter counter measurements, bacterial culture volumes ranging from 10 to 100 µL were diluted in 12 mL of Isotone II diluent in a Coulter counter cuvette (Beckman Coulter, 580015). The diluted samples were then vortexed at a medium speed. After vortexing, the samples

were transferred to a Multisizer 3 Coulter counter (Beckman Coulter), equipped with a 10 μm aperture. The samples were allowed to stabilize for 1–2 min before initiating the measurement. In each measurement, a 500-μL sample is analyzed by the device.

## Spot assay

Plates for the spot assay were prepared using the DAOA technique (see Growth media). First, the underlay medium was microwaved until liquefied and then cooled to 60 °C, where it was held overnight. Subsequently, 38 mL of this medium was poured into an Nunc OmniTray single-well plate (Thermo Fisher Scientific, 242811) and left to dry under a laminar flow hood for 15 min. For the overlay, bacterial overnight culture was added to the overlay medium at 60 °C, to achieve an OD of 0.4. In total, 5 mL of this mixture was then layered over the solidified underlay in the omnitray plate. Finally, phage samples were diluted in a 1:10 dilution series and 5 μL droplets from each dilution ($10^{-5}$ to $10^{-9}$) were carefully applied to the overlay using a MINI 96 electronic 96-channel pipette (Integra, 4803). Plaque-forming unit (PFU) counting was conducted manually for all dilutions where plaques did not overlap.

## Statistics

Statistical tests (Pearson, Kendall rank, and Wilcoxon signed rank) were implemented using Python Scipy.stats (v1.11.1, resp. pearsonr, kendalltau and willcoxon). For multiple testing correction, we used Python statsmodels (v0.14.0). Bootstrapping [44] was performed by randomly resampling the time series 1,000 times using a custom Python script before extracting the collapse times.

## Analysis of bacterial growth curves

The bacterial growth rate was extracted from bioluminescence curves of bacterial cultures without phages. Curves were fitted in the range of $t > 0.5$ h and bioluminescence $< 10^6$ AU. A first-order polynomial was fitted (Levenberg–Marquardt method implemented with Python's optimize.curve_fit, SciPy 1.11.1) to the bioluminescence values in log space to ensure that the low concentration part of the curve also contributed to the fit. The slope of the curve was taken as the bacterial growth rate, whereas the intersection was used to estimate the initial bacterial concentration (see below). The collapse time of phage predation curves was extracted by using Python's find_peaks (SciPy 1.11.1), using a width of four data points (20 min) for smoothing. The analysis script can be found in https://doi.org/10.5281/zenodo.14801073. Collapse points with bioluminescence values above $3 \times 10^5$ AU were excluded, as they were outside of the steady-state exponential growth regime. When multiple initial conditions were present of the same phage in the same growth conditions (Figs 2d, 3c and 3d, and S7), Eq 2 was fitted using optimize.curve_fit, SciPy 1.11.1 to extract the phage amplification rate from multiple growth curves. Otherwise, the phage amplification rate was calculated from a single growth curve using:

$$r_{bac} = \frac{\log\left(p_\infty / p_0\right)}{b_0 \exp\left(t_{col} r_{bac}\right) - b_0} \tag{3}$$

$r_{bac}$ was extracted from a bacterial growth curve in the absence of the phage, as described above. For phage amplification rates in the presence of an antibiotic (e.g., Fig 4), we measured $r_{bac}$ in the presence of the antibiotic at the relevant concentration, without any phage. $b_0$ was measured using a Coulter counter. We determined the ratio $p_\infty / p_0$ in a high-throughput manner inspired by the study [9]: We purified the phage from the experiment of interest

(e.g., in the presence of antibiotic), diluted it by a factor of $d$ (typically 60–300) and added bacteria from a fresh stock to measure the collapse time. For each phage, we create a calibration curve by measuring the collapse time at various phage stock dilutions ( $p_{stock} / p_0$ ) against the same fresh bacterial stock. We fit the calibration curve with the following equation:

$$t_{col,stock} = a \log\left( p_{stock} / p_0 \right) + b \tag{4}$$

We then extract the relative phage concentration using:

$$\frac{p_\infty}{p_{stock}} = e^{(t_{col} - b)/a} d \tag{5}$$

Lastly, by multiplying $p_\infty / p_{stock}$ with the dilution $d$ of the first growth experiment ($p_{stock}$/$p_0$), we obtain $p_\infty / p_0$.

The standard errors of the phage amplification rate (Figs 1c, 4b) and the phage concentration (Fig 4c) were determined by bootstrapping the growth curves 1,000× and determining the collapse time to estimate their standard deviations. This error was then propagated to either the phage amplification rate or the relative phage stock concentration (Eq 3 or 5, respectively).

Notably, the phage amplification rate only logarithmically depends on $p_\infty / p_{stock}$. Since the experimental determination of $p_\infty / p_0$ is relatively laborious to perform for many phages simultaneously, even with the above method, we tested whether the phage amplification rate could be approximated by assuming $p_\infty / p_0 \approx p_{stock} / p_0$. We tested this for nine different phages (S9c Fig) and found that this assumption gives very similar results to explicitly measuring $p_\infty / p_0$ (Pearson's $\rho = 0.95$, $p = 10^{-22}$). Therefore, when measuring the amplification rate of the entire BASEL collection (Fig 3), we omitted the measurement of $p_\infty / p_0$ and instead assumed $p_\infty / p_0 = p_{stock} / p_0$.

For the measurements of antibiotic-dependent phage amplification rates (Figs 4 and S9) we had many different initial conditions. While plate reader experiments can be performed in high throughput on 96-well plates, measuring the initial bacterial concentration ( $b_0$ ) required single sample Coulter counter experiments. Therefore, we measured to what extent the initial bioluminescence intensity of growth curves (see above) could replace the Coulter counter experiments. For a limited set of replicates (1 × 8 concentrations of nitrofurantoin and 2 × 8 concentrations of doxycycline) and nine different phages, we measured the phage amplification rate by measuring $b_0$ with the Coulter counter and $p_\infty / p_0$ with subsequent bacterial growth curves. We compared the amplification rate determined from both of these measured parameters (abscissa, S9c Fig) with the amplification rate where $b_0$ is semi-quantitatively approximated by the extrapolated bioluminescence intensity at $t = 0$ and approximated $p_\infty / p_0 \approx p_{stock} / p_0$ (ordinate, S9c Fig). We found a clear correlation between both quantities (Pearson $r = 0.67$, $p = 10^{-44}$) and therefore performed more replicate experiments (five per condition) using the much less laborious approximation of phage amplification rate (x-axis) for full characterization (Fig 4b, 4d). It should be noted that while the Coulter counter provides a quantitative readout of bacterial concentration, bioluminescence provides only a semi-quantitative readout of bacterial concentration. However, this is sufficient for measuring antibiotic-dependent changes in amplification rate.

## Model of phage amplification kinetics

### Adsorption-limited regime

Phage amplification kinetics are modeled as a two-step process: Phages must first adsorb to bacteria at a rate $k_{adsorb}$, after which they cause lysis of bacteria after time $\tau_{latency}$ (latency

period). With each cycle, $n_{\text{burst}}$ phages are created and one bacterium disappears. Note that the burst size is a composite parameter of the product of the number of phages released multiplied by the frequency of replicative progeny (e.g., for phages with an efficiency of plating below 1). We first consider the adsorption-limited regime ($k_{\text{adsorb}} b \ll 1 / \tau_{\text{latency}}$). The condition for the collapse time of the bacterial population is that the bacterial death rate due to the phage is equal to the growth rate of the bacteria, which implies that the collapse time is implicitly given by:

$$p(t_{\text{col}}) = \frac{r_{\text{bac}}}{k_{\text{adsorb}}} \tag{6}$$

Next, we calculate the collapse time of the bacterial population by neglecting the phage contribution in (Eq 1a). In this approximation, $b(t) = b_0 e^{r_{\text{bac}} t}$.

For a known initial bacterial concentration $b_0$, we can obtain the phage concentration as a function of time by integrating (Eq 1b). In this regime, the phage population grows super-exponentially from its initial value $p_0$:

$$p(t) = p_0 e^{\frac{n_{\text{burst}}}{r_{\text{bac}}} k_{\text{adsorb}} b_0 (\exp(r_{\text{bac}} t)-1)} \tag{7}$$

We then insert in the collapse condition (Eq 6) and solve for $t_{\text{col}}$, to obtain

$$t_{\text{col}} = \frac{1}{r_{\text{bac}}} \log\left(1 + \frac{r_{\text{bac}}}{k_{\text{adsorb}} n_{\text{burst}} b_0} \left(\log\left(\frac{r_{\text{bac}}}{p_0 k_{\text{adsorb}}}\right)\right)\right) \tag{8}$$

We define the phage amplification rate as $r_{\text{pha}} = n_{\text{burst}} k_{\text{adsorb}}$ and eliminate the dependence on $k_{\text{adsorb}}$ by approximating the phage concentration at the onset of collapse as $p(t_{\text{col}}) = \frac{r_{\text{bac}}}{k_{\text{adsorb}}} \approx p_\infty$, with the final phage concentration $p_\infty$ (phage yield). We have validated his approximation experimentally (S10 Fig). Using this approximation, we obtain the collapse time in (Eq 2):

$$t_{\text{col}} = \frac{1}{r_{\text{bac}}} \log\left(1 + \frac{r_{\text{bac}}}{r_{\text{pha}} b_0} \log\left(\frac{p_\infty}{p_0}\right)\right) \tag{9}$$

Finally, we use the measured collapse time to infer the phage amplification rate by solving Eq 9 for $r_{\text{pha}}$:

$$r_{\text{pha}} = \frac{r_{\text{bac}}}{b_0 \left(e^{r_{\text{bac}} t_{\text{col}}} - 1\right)} \log\left(\frac{p_\infty}{p_0}\right) \tag{10}$$

We find good agreement between this model and the experimental data (Figs 2d, 3c).

## Latency-limited regime

So far, we have considered the adsorption-limited regime where $k_{\text{adsorb}} b(t) \ll 1 / \tau_{\text{latency}}$. We now consider the opposite regime, where latency bursting is limiting:

$$\frac{db}{dt} = r_{\text{bac}} b(t) - p(t) / \tau_{\text{latency}} \tag{11}$$

$$\frac{dp}{dt} = p(t) n_{\text{burst}} / \tau_{\text{latency}} \tag{12}$$

In this regime, the dynamics can be solved analytically, yielding

$$b(t) = b_0 e^{r_{\text{bac}} t} - \frac{p_0 \left( e^{n_{\text{burst}} t / \tau_{\text{latency}}} - e^{r_{\text{bac}} t} \right)}{n_{\text{burst}} - r_{\text{bac}} / \tau_{\text{latency}}} \tag{13}$$

$$p(t) = p_0 e^{n_{\text{burst}} t / \tau_{\text{latency}}} \tag{14}$$

Next, we calculate the collapse time ( $db/dt\big|_{t=t_{\text{col}}} = 0$ ). We find that the population collapses at a fixed MOI ( $\text{MOI} = p/b$ ):

$$p_{\text{col}} / b_{\text{col}} = \text{MOI}_{\text{col}} = r_{\text{bac}} \tau_{\text{latency}} \tag{15}$$

Imposing the collapse condition, we obtain for the collapse time:

$$t_{\text{col}} = \frac{\log\left( r_{\text{bac}} \tau_{\text{latency}} \left( \frac{1}{n_{\text{burst}}} + \frac{1}{\text{MOI}_0} - r_{\text{bac}} \tau_{\text{latency}} \right) \right)}{\dfrac{n_{\text{burst}}}{\tau_{\text{latency}}} - r_{\text{bac}}} \tag{16}$$

Therefore, we find that the collapse time in the latency-limited regime depends only on the initial MOI and not on the absolute phage and bacterial concentrations separately. This result is inconsistent with the experimental results in Figs 1c and S7a, as both show a monotonic decrease in collapse time as a function of the initial bacterial concentration at a fixed initial MOI. In principle, the experimental system could be in the intermediate regime where $k_{\text{adsorb}} b(t)$ is on the order of $1 / \tau_{\text{latency}}$ for at least for some duration of the experiment. However, our experimental results indicate that this situation can only occur relatively late (if at all), in the last few infection cycles before the collapse. The fact that the adsorption-limited case (Eq 9) already shows good agreement with the experimental data (Figs 2d, 3c) supports that it captures the essence of the dynamics up to the collapse and is thus sufficient as a minimal model. Therefore, we do not consider more complex models with inherently more free parameters.

## Comparison of full growth curves between model and experiments

We further validated the model dynamics by comparing it with the experimentally measured bacterial growth curves in Fig 2. This requires a joint estimate of $r_{\text{pha}}$ and $k_{\text{adsorb}}$. We obtain an estimate of $k_{\text{adsorb}}$ by taking advantage of the high degree of variation in the initial conditions, noting that a high initial phage concentration must imply a fitness cost for the bacterial population. Indeed, the ratio of the bacterial concentration at the time of collapse to the initial bacterial concentration depends on the initial phage concentration (S10 Fig). We then computed $r_{\text{pha}}$ using Eq 2 for each initial condition and took the average over the initial conditions. Using $k_{\text{adsorb}}$ and $r_{\text{pha}}$, we integrated Eq. 1a, 1b. We integrated the dynamics using the 10th data point as the initial condition to avoid initial noise in the experimental data.

## Model extension including resistance and lysogeny

To investigate the effect of resistance and lysogeny on bacterial growth curves, we extended the mathematical model (S4 Fig). First, we described the number of bacteria sensitive to phage infection:

$$\frac{\mathrm{d}b_{\mathrm{sens}}}{\mathrm{d}t} = b_{\mathrm{sens}}(t)(r_{\mathrm{bac}} - r_{\mathrm{res}}) - p(t)k_{\mathrm{adsorb}}b_{\mathrm{sens}}(t) \tag{17}$$

where $r_{\mathrm{res}}$ is the rate at which sensitive bacteria evolve phage resistance. For simplicity, we assume that resistant bacteria grow at the same rate as sensitive bacteria and are completely unaffected by the phage:

$$\frac{\mathrm{d}b_{\mathrm{res}}}{\mathrm{d}t} = r_{\mathrm{bac}}b_{\mathrm{res}}(t) + r_{\mathrm{res}}b_{\mathrm{sens}}(t) \tag{18}$$

Finally, we included lysogeny by assuming that a fraction of the bacteria, $\phi_{\mathrm{lys}}$, is lysogenized:

$$\frac{\mathrm{d}p}{\mathrm{d}t} = p(t)n_{\mathrm{burst}}k_{\mathrm{adsorb}}b_{\mathrm{sens}}(t)(1 - \phi_{\mathrm{lys}}) \tag{19}$$

The lysogenized bacteria are insensitive to further phage predation, and for simplicity, we assume that they grow at the same rate as sensitive bacteria:

$$\frac{\mathrm{d}b_{\mathrm{lys}}}{\mathrm{d}t} = r_{\mathrm{bac}}b_{\mathrm{lys}}(t) + p(t)k_{\mathrm{adsorb}}\phi_{\mathrm{lys}}b_{\mathrm{sens}}(t) \tag{20}$$

Note that we retrieve the limiting cases of no resistance evolution and no lysogeny with $r_{res} = 0$, respectively.

## Supporting information

**S1 Text. Requirements to determine the phage amplification rate.**
(PDF)

**S1 Fig. Comparison of collapse times obtained from bioluminescence and optical density measurements. (a)** Example of an *E. coli* growth curve in the absence (blue) and presence (orange) of a phage (Bas04), as measured by optical density (Methods). Initially, there was condensation on the lid of the 96-well plate, resulting in an artifactually higher optical density for the first half hour. **(b)** The collapse time of phage Bas04 measured with both bioluminescence and optical density for the same samples. All combinations of seven MOIs (0.0004–0.3, 3× steps) and 11 bacterial concentrations ($3 \times 10^3$–$3 \times 10^6$, 2× steps) were measured. Both methods show close agreement (Pearson's $\rho = 0.99$, $p = 10^{-65}$). The data underlying this figure can be found in S1 Data.
(PDF)

**S2 Fig. Calculation of the phage concentration from growth curves. (a)** Bacterial growth curves were measured in the presence of several different dilutions (1/800, 1/1,600, and 1/3,200, respectively, orange, pink, and blue) of the Bas04 phage stock used in Figs 1 and 2 at a fixed initial bacterial concentration ($4 \times 10^5$ mL$^{-1}$) to quantify the phage concentration-dependent collapse time. **(b)** We fit these collapse times ($t_{\mathrm{col}}$) with a logarithmic decay (black line, see Ref. [9] for details) to obtain a calibration curve that we use to determine unknown phage concentrations in samples containing the phage Bas04. The data underlying this figure can be found in S1 Data.
(PDF)

**S3 Fig. Comparison of full growth curves between model and experiments.** Comparison between measured bacterial growth curves (black) and model trajectories (red, dotted) as a function of the different initial bacterial concentrations ($1.1 \times 10^4$–$1.1 \times 10^6$ mL$^{-1}$, left to right) and 10 different initial phage concentrations ($2.5 \times 10^3$–$1.3 \times 10^6$ mL$^{-1}$, top to bottom). The procedure for calculating the model trajectories is explained in section *Comparison of full growth curves between model and experiments*. Overall, the model trajectories agree well with the measured bacterial growth curves up to collapse and capture the collapse time. The quality of the fit depends on the initial conditions: The higher the initial bacterial concentration, the further the system is from the adsorption-limited regime, and the worse the agreement with the model. The experimental data underlying this figure can be found in S1 Data.
(PDF)

**S4 Fig. Resistance and lysogeny affect only post-collapse dynamics.** Results of a mathematical model including phage resistance and lysogeny (Eqs. 17–20). When resistance (orange dashed line) and lysogeny (green dashed line) are included, the dynamics up to the collapse time and including the initial collapse are not noticeably affected. The curve for resistance and lysogeny is not shown because it is indistinguishable from the curve for lysogeny alone. For the simulations, we used an initial bacterial density of $10^5$ cfu/mL, an initial phage density of $10^5$ pfu/mL, a bacterial growth rate of $1\,h^{-1}$, a resistance frequency of $10^{-8}$, an adsorption rate of $10^{-9}$ cfu/mL/h, a burst size of 100, and a lysogeny frequency of $10^{-6}$, which are all realistic parameters [9]. The *y*-axis shows the total bacterial population (sum of the sensitive, resistant, and lysogenized populations).
(PDF)

**S5 Fig. PHORCE also captures the collapse time for a *Pseudomonas aeruginosa* strain and its phage.** We measured growth curves of a *P. aeruginosa* clinical isolate in the presence of a therapeutically used phage (Methods) to test whether PHORCE also works for bacteria other than *E. coli* laboratory strains. (**a**) Bacterial growth curves from a two-dimensional gradient of initial bacterial densities (serially diluted by a factor of 3, from left to right: 460–$3.3 \times 10^5$ cfu/mL) and initial phage densities (serially diluted by a factor of 3, from black to red: 30–$1.7 \times 10^6$ pfu/mL). (**b**) Collapse time versus initial bacterial density for the experiment in a. Similar to our results for *E. coli* (Fig 1c), the collapse time decreases logarithmically with the bacterial density for a fixed initial bacterial-to-phage ratio (CFU/PFU = 0.2). (**c**) The PHORCE model (Eq 1) quantitatively captures the dependence of the collapse time on the initial bacterial and phage densities, using the phage amplification rate as the only free parameter. The fitted phage amplification rate was $1.4 \times 10^{-6}$ mL h$^{-1}$, which is comparable to the strongest BASEL phage. Such a high rate is plausible because this phage was selected for its strong activity. The data underlying this figure can be found in S1 Data.
(PDF)

**S6 Fig. Raw bacterial growth curves in the presence of the BASEL phages.** Each graph shows a time-resolved bacterial growth curve, measured using bioluminescence (Methods), in the presence of a different phage. Green to black shows different initial bacterial concentrations (resp. 480, $1.44 \times 10^4$ and $4.32 \times 10^5$ mL$^{-1}$) at a fixed initial MOI, and red to black shows different phage concentrations at a fixed initial bacterial concentration (resp. $8.1 \times 10^5$, $2.7 \times 10^4$ and 900× diluted from the phage stock solutions). The first column shows the BASEL phages 1–12, the second column shows 13–24, and so on (top to bottom). The data underlying this figure can be found in S1 Data.
(PDF)

**S7 Fig. Phage amplification characteristics of the BASEL collection.** Analysis of the growth curves in S6 Fig. (a) The collapse time was measured at three bacterial densities, keeping the ratio of phages to bacteria (multiplicity of infection [MOI]) fixed, for the entire BASEL collection. $t_{col,1-3}$ indicate time of collapse at resp. the lowest, middle and highest bacterial density. Each data point represents a different phage. All data points below the two solid lines show a monotonic increase in collapse time as a function of the bacterial concentration. The dashed line indicates where the collapse time increases with the logarithm of the bacterial concentration. The vast majority of data points lie below the two solid lines and cluster around the dashed line, suggesting that the amplification kinetics of all BASEL phages are predominantly adsorption-limited. (b) Phage amplification rates for different phage morphotypes, 3/6 podoviruses, 0/34 siphoviruses, and 8/29 myoviruses had a phage amplification rate below the detection limit ($3 \times 10^{-10}$ mL h$^{-1}$). None of the morphotypes differ significantly from the others ($p = 0.08$, Kruskal–Wallis) and any differences become even more insignificant if the vequintavirus members (Fig 3e) are excluded from the myoviruses ($p = 0.87$, Kruskal–Wallis). (c) Comparison between the overnight yield ($y$-axis) and the phage amplification rate ($x$-axis) for each phage in the BASEL collection shows no significant correlation ($\tau = -0.04$, $p = 0.6$, Kendall rank correlation). (d) Comparison between the phage amplification rate ($x$-axis) and its approximation by assuming $p_\infty / p_0 = p_{stock} / p_0$ ($y$-axis). Each data point shows the phage amplification rate determined from a single amplification curve (nine phages, three replicates per condition). Both results are in close agreement (Pearson's $\rho = 0.95$, $p = 10^{-22}$). The data underlying this figure can be found in S1 Data.
(PDF)

**S8 Fig. Bacterial growth curves in the presence of antibiotics and phages.** Bioluminescence as a function of time. Each row shows a different phage from the BASEL collection (phages from top to bottom: Bas01, Bas04, Bas10, Bas14. Bas17, Bas23, Bas30, Bas37, Bas42, Bas55, and Bas63); each column shows a different antibiotic (left: nitrofurantoin, orange; right, doxycycline, blue) at different concentrations ($0.02$–$0.2 \times$ MIC from dark to colored, where black is no antibiotic). The data underlying this figure can be found in S1 Data.
(PDF)

**S9 Fig. Phage–antibiotic interactions.** (a) The degree of correlation between the phage amplification rate and the antibiotic concentration was assessed using either a Kendall correlation test ($x$-axis) or a Pearson correlation test ($y$-axis). Both scores are significantly correlated with each other (Kendall's $\tau = 0.5$, $p = 0.002$). (b) No correlation was found between the interaction score based on the phage yield and the interaction score based on the amplification rate ($p = 0.6$, Kendall rank correlation). (c) The phage amplification rate was calculated either by explicitly measuring the initial bacterial concentration and the final phage concentration ($x$-axis) or by approximating the initial bacterial concentration with the initial bioluminescence value and assuming $p_\infty / p_0 = p_{stock} / p_0$ ($y$-axis) (Methods). The explicit and approximate approaches yield similar results (Pearson's $\rho = 0.67$, $p = 10^{-44}$). On both axes, the antibiotic-dependent phage amplification rate was normalized to the phage amplification rate in the absence of the drug. For both doxycycline and nitrofurantoin, nine phages and eight drug concentrations were measured, respectively. The data underlying this figure can be found in S1 Data.
(PDF)

**S10 Fig. Dependence of phage yield on time of measurement.** Results of an experiment with a two-dimensional gradient of initial phage (Bas14) and bacterial concentration, similar to Fig 2. (a) Bacterial growth curves in the two-dimensional gradient. Different columns show different initial bacterial densities (left to right: 1:100–1:12,800 dilutions from a 1:100 diluted

overnight culture pre-cultured for 2 h using seven steps of 5-fold serial dilutions); different colors show different initial phage concentrations (black to red: $0.001–6.25 \times 10^{-5}$ relative to the stock concentration using four steps of 5-fold serial dilutions). (**b**) Phage yield versus collapse time. We took phage samples at $t = 8$ h (blue) and at $t = 21$ h (orange) and measured the relative phage concentration by reinoculation (S2 Fig). At these time points, the bioluminescence drops slightly as the cultures are removed from the plate reader and therefore cool down slightly before being returned to the temperature-controlled plate reader. Since the samples have different initial conditions, these yields (blue data points) correspond to different time points relative to the collapse time: The $x$-axis shows the time of the first phage concentration measurement ($t = 8$ h) minus the collapse time. (**c**) Ratio of phage amplification rate calculated from the phage yields measured after 8 h and after 21 h. The data underlying this figure can be found in S1 Data.
(PDF)

**S1 Table. BASEL collection metadata.** The table lists all details regarding the taxonomic classification, isolation/source, host receptors, and genomic features of all *E. coli* phages used in this study. The data were taken from Ref. [20] and complemented with the phage amplification rates and phage stock concentrations characterized in this work.
(XLSX)

**S1 Data. Numerical experimental data for all figures.**
(XLSX)

## Acknowledgments

We thank Alexander Harms for sharing the BASEL collection, Jean-Paul Pirnay for sharing the Pseudomonas phage and Antiinfective Intelligence for sharing the Pseudomonas clinical isolate; Gerrit Ansmann, Theresa Fink, Michael Lässig, and Rob Lavigne for feedback on the manuscript; and the entire Bollenbach group for fruitful discussions.

## Author contributions

**Conceptualization:** Yuval Mulla, Tobias Bollenbach.

**Formal analysis:** Yuval Mulla, Denny Trimcev.

**Funding acquisition:** Tobias Bollenbach.

**Investigation:** Yuval Mulla, Janina Müller, Denny Trimcev.

**Supervision:** Tobias Bollenbach.

**Validation:** Denny Trimcev.

**Visualization:** Janina Müller.

**Writing – original draft:** Yuval Mulla.

**Writing – review and editing:** Yuval Mulla, Janina Müller, Denny Trimcev, Tobias Bollenbach.

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
