## [Editor Report · Decision Letter 0]

10 Jul 2024

Dear Tobias,

Thank you for submitting your manuscript entitled "Extreme diversity of phage amplification rates and phage-antibiotic interactions revealed by PHORCE" for consideration as a Short Report by PLOS Biology.

Your manuscript has now been evaluated by the PLOS Biology editorial staff, as well as by an academic editor with relevant expertise, and I am writing to let you know that we would like to send your submission out for external peer review.

Once your full submission is complete, your paper will undergo a series of checks in preparation for peer review. After your manuscript has passed the checks it will be sent out for review. To provide the metadata for your submission, please Login to Editorial Manager (https://www.editorialmanager.com/pbiology) within two working days, i.e. by Jul 12 2024 11:59PM.

Kind regards,

Melissa

Melissa Vazquez Hernandez, Ph.D.

Associate Editor

PLOS Biology

---

## [Decision Letter · Decision Letter 1]

2 Sep 2024

Dear Tobias,

Thank you for your patience while your manuscript "Extreme diversity of phage amplification rates and phage-antibiotic interactions revealed by PHORCE" was peer-reviewed at PLOS Biology. It has now been evaluated by the PLOS Biology editors, an Academic Editor with relevant expertise, and by three independent reviewers. I again would like to apologize for the long review process.

In light of the reviews, which you will find at the end of this email, we would like to invite you to revise the work to thoroughly address the reviewers' reports. As you will see below, majority of reviewers are positive about the relevance and novelty of the study, yet some concerns have raised during revision. Reviewer #1 wonders how the model's accuracy might be influenced by the precision of cell growth measurements and phage yield. In addition, reviewer #2 suggests evaluating the predictive ability of the phage amplification rate in another simple model of phage therapy, fitting population trajectories to the model, and simulating different phage lifestyles. Reviewer #3 requests to explore whether it can be used beyond E. coli and to discuss limitations related to phage yield and the stability of the amplification rate in different culture conditions. In addition, reviewers #1 and #3 think the methods should be more clear. Although we do not expect that you fully address all reviewer comments with new experimental work, we do require additional validation efforts by experimental validation of the phage-host work (as suggested by reviewer #1 and #3) along with some further simulations (as suggested by reviewer #2).

Given the extent of revision needed, we cannot make a decision about publication until we have seen the revised manuscript and your response to the reviewers' comments. Your revised manuscript is likely to be sent for further evaluation by all or a subset of the reviewers.

**IMPORTANT - SUBMITTING YOUR REVISION**

*Re-submission Checklist*

*Published Peer Review*

*PLOS Data Policy*

*Blot and Gel Data Policy*

Sincerely,

Melissa

Melissa Vazquez Hernandez, Ph.D.

Associate Editor

PLOS Biology

REVIEWERS' COMMENTS:

Reviewer #1:

Mulla et al. present an exciting new method to measure phage amplification rates in a high-throughput manner. They show that phage infection kinetics under assay conditions seem to be adsorption-limited and derive an analytical expression for the collapse time of the bacterial population. They use their method to show the wide diversity of phage amplification rates, which do not correlate with phage morphology. They also find that interactions between phages and antibiotic are determined largely by the antibiotic and not the phage species.

Overall, the paper presents a valuable method that could alleviate a current limitation in phage research. The authors underline this by presenting novel insights into phage amplification and phage-antibiotic interactions. My comments mainly concern unclarities about the method and presentation of its output.

1) Model: How dependent is the model on an accurate estimate of bacterial growth rate? At least for optical density measurements, growth rate estimates might be affected by changes in cell size, e.g. through filamentation in the presence of antibiotics or between different media conditions.

2) Plate preparation: the dilutions are described in a relatively convoluted way in the section 'Two-dimensional gradient of initial phage and bacteria concentration', which does not particularly lend itself to repeating the method. It is also unclear how the double gradient was achieved as it sounds like first the phage gradient was pipetted and then the bacterial gradient was also done by dilution. Would that not change the phage concentration in the wells and how was this calculated?

It would be helpful to have a schematic of how the experiments were performed to make reproduction and understanding easier.

3) I agree with the authors that the phage amplification rate is a very useful measurement of phage efficacy, which might be similar to the bactericidal effect of antibiotics. Yet, as can be seen in many growth curve figures (e.g. Fig. S5), in many cases the repression of bacterial numbers is quite transient and this rebound is obviously an important characteristic for applications like phage therapy as well. It would be good to at least discuss this more. Particularly as this might be the reason that final phage yields are not correlated with phage amplification.

4) Related to the last point, phage yield was used to approximate the phage concentration at the onset of collapse (SI), but phage yield was measured after ON incubation, which is in most cases many hours later than collapse time and it is not quite clear how accurate this approximation is or if it matters.

5) Similarly, the authors approximated p_inf/p_0 by p_stock/p_0 and b0 from bioluminescence for some of the experiments. While they tested the validity of this approximation, it is buried in the Methods. It would be good to include this in a sketch of the method to make it clear, which measurements are necessary (or can be simplified) and how they go into the determination of the amplification rate.

Out of curiosity: It seems that in Fig. S6c, the approximations lead to better amplification rate estimates with NIT than DOX. Is that coincidence or something to do with how the antibiotics influence phage amplification?

6) The authors use Kendall rank correlation, Pearson correlation and Kruskal-Wallis test for their data but there does not seem to be a statistics section in the Methods, summarizing this data analysis.

7) The Methods mention the use of frozen stocks of overnight cultures, which seems a bit unusual, why were not fresh ONCs used?

8) It is interesting that one phage seemed to have neutral interaction with both antibiotics. Do you have an hypothesis what makes this phage different?

Reviewer #2:

The authors set out to build a rigorous mathematical framework and experimental approach to quantify phage activity. They argue that existing experimental methods are error-prone, often ad hoc, and limited by their dependence on difficult-to-acquire phage population dynamic measurements. I agree that the authors are addressing a highly pressing question, as a better quantitative grasp of phage activity is needed to better design phage therapies.

The core of the author's PHORCE method is an expression relating the phage amplification rate, essentially the per-capita growth rate of the phage, to the collapse time of bacterial population. To derive this, the authors define a simple mathematical model of a lytic phage and assume the dynamics are adsorption limited, which they justify by arguing their data is inconsistent with a burst-limited model. In their model, the phage amplification rate is the product of the adsorption rate and the burst size. They show that collapse time with phage Bas04 shows a logarithmic decrease with initial bacterial abundance, consistent with their expression. The authors then perform a series of experiments with Bas04 at varying initial phage and bacterial concentrations, finding all the collapse times can be explained by fitting a single value of amplification rate.

The authors then apply their method to the BASEL collection of lytic E. coli phages, finding that amplification rates varying across orders of magnitude. As validation, they show that the expected collapse time scaling is generally conserved in this collection. They find a minimal relationship between amplification time and phage yield. Finally, they apply their method to combination treatments of phage and antibiotic. They find that, across multiple phages, trends in phage-antibiotic interaction tend to be determined by the antibiotic, with multiple phages having similar interaction patterns with the same antibiotic.

Overall, I think this manuscript is a worthwhile contribution to the literature. It establishes a more rigorous framework for understanding the dynamical impact of phage on bacteria, akin to MIC's role in understanding antibiotics. However, I have concerns about the generality of the author's mathematical framework and phage amplification metric.

Major comments

1. The authors center their manuscript around the composite phage amplification rate parameter, but do not explore how meaningful this quantity is for predicting phage dynamics beyond collapse time. To what extent does this quantity uniquely define phage dynamics? Can there be substantial difference in the ability of a phage to clear an infection among phage with the same amplification rate? Further simulation work exploring the predictive ability of phage amplification rate in a simple mathematical model of phage therapy would improve the manuscript.

2. The authors show that their model is able to explain the scaling of the collapse time, but is their minimal model able to quantitatively fit the population trajectories of the bacteria? If the model can explain the collapse scaling, but not the trajectory, what is the significance of this deviation? Does it indicate that the amplification rate measured via collapse time is a temporal average? This can be addressed by attempting to fit the model to trajectories and further simulation with modified phage models.

3. The authors consider only a narrow range of possible phage lifestyles, but there is no guarantee that the phage isolated for future phage therapy efforts will obey these constraints. For example, little mention of lysogenic phage is made, how will the accuracy of the estimates be altered if the method is applied to a partially lysogenic phage? The authors should make more extensive simulation of other phage lifestyles to see how their estimates are affected.

Minor comments

1. Given the importance of the model in this manuscript, showing the ODEs in the main text would be worthwhile.

2. It is currently unclear how the bacterial growth rate parameter in Eq. 1 is handled in the phage-antibiotic interaction experiments. Did the authors measure bacterial growth rate in the presence of the antibiotic concentration? Or use a growth rate from a no-antibiotic control?

3. SI Fig 4b may be better suited as a main text figure. Validating that the collapse scaling still works in other species is an extremely important verification step.

4. Given that the authors only tested two antibiotics, I feel they should soften the generality of their claims on the antibiotics determine phage-antibiotic interactions.

5. Can this method potentially be adapted to study phage-phage interactions?

Reviewer #3:

Review "Extreme diversity of phage amplification rates and phage-antibiotic interactions revealed by PHORCE" by Mulla et al.

The manuscript "Extreme diversity of phage amplification rates and phage-antibiotic interactions revealed by PHORCE" by Mulla et al. presents a novel method to directly quantify the bactericidal effect of bacteriophages. Such a method provides important complementary information to traditional phage parameters such as burst size and latent period, and for modeling purposes could eliminate the need to measure these traits, which are elaborate and time-consuming. The manuscript combines experiments and mathematical modeling work. Experiments were performed meticulously and the results seem promising. Since this method would be desirable for enhancing and accelerating quantitative phage research, including fundamental research as well as therapeutic use of phages, we believe that this manuscript can have a great impact in the phage field. We would like to give the authors the following points for consideration. Please mind that you use always line numbers in submitted manuscripts.

Major points:

1. P12 L18, L27, L32, L37 and also in the Discussion, you several times mention "under our conditions" and "adsorption-limited regime". Can you say a few words about the types of phages where this could be different, and how that would impact the approach/results? The finding that the system is adsorption-limited is counter-intuitive. I could only find a brief statement on P13: "While the burst rate may also contribute to the dynamics, assuming adsorption as the sole rate-limiting step already quantitatively explains our results." Still it remains unclear in which cases this might not hold. Could the authors speculate?

2. P13 L42: you mention that "some phage amplification rates were below the detection limit of our technique" - please elaborate, what does that mean? How often does that happen? In the Fig3 caption I found the statement that "slower growing phages did not cause a collapse within the exponential growth regime of the bacteria", but this point should receive more attention in the main text. How common can we expect that to be for other phages, e.g. of faster or slower growing hosts, and/or for other hosts than E. coli? This is important, as it is a caveat that precludes using PHORCE for some bacteria, and this might be very common (we rarely observe collapse events and have worked with many different phages and bacteria). Can the authors provide a directive as to how this issue can be resolved for such cases? This is especially important since the introduction gives off the impression that the method is universally applicable. If only data for E. coli is shown, the paper should strictly be reframed as providing a method that is suitable for E. coli and still needs to be tested on other bacteria, but I realize that would be a pity.

3. P14 L22: I'm surprised that "phage yield did not correlate at all with this key quantity (Suppl. Fig. 4d)." Can you give an intuitive explanation? Could you reflect on whether this could be due to the alternative approach of phage particle counting? What about other parameters? In the BASEL phage collection paper they also measured the efficacy of plating (EOP), which might be used as an indication of host range. Maybe you could correlate this with the bactericidal effect to address the "fundamental question" on P17 L21?

4. P15 L44-45: regarding the statement that "the phage yield decreases similarly with increasing concentration for both antibiotics (Fig. 4d)" and "In contrast to the phage amplification rate, the yield is not markedly different between the two drugs" (Fig4 caption). It looks like the yield drops for NIT, how significant is that? Would you discuss what could be an explanation (maybe quicker release with lower burst size)? Also, the Y axis of Fig 4d is inconveniently cut at the low end, please fix this. While in Suppl fig 6b, there is no general trend between the interaction scores, the specific cases represented by outliers on the X and Y axes may still be interesting to note and/or investigate further?

5. P16 L2-5: "Together with the lack of correlation between phage amplification rate and phage yield across the entire BASEL collection (Suppl. Fig. 4d), these results show that the phage amplification rate is a distinct property of phages that is largely independent of the commonly used phage yield." From this statement, the parameter should be termed differently, e.g. "bactericidal effect" or "virulence". This is because the "phage amplification rate" is intuitively connected to phage growth, but this parameter actually reflects the ability of the phage to inhibit bacterial growth.

6. P16 L21: "directly compared across different phages, hosts, and growth conditions" but two lines later you say: "hard to standardize across experiments". How stable is the phage amplification rate for different culturing conditions? If we want to compare phages from different studies and systems, experimental conditions are bound to differ, so this will impact conclusions and interpretation of more complex situations. Please address this potential caveat in sufficient detail.

7. P18 L32, P19 L12: I really like that the methods for "Two-dimensional gradient of initial phage and bacterial concentration" and "Antibiotic gradients" are thoroughly described, but they are quite difficult to follow and it is difficult to visualize the experiment from the text alone. For example, "Subsequently, a gradient of phage titers was established in a 12-column deep well plate using a 1:2 serial dilution procedure, resulting in 6 ml per phage concentration." It is hard to understand how the 6 ml per phage concentration comes about. An addition of figures or organizing the steps would make it easier to understand.

Minor points:

8. Please remove the word "Extreme" from the title, as this is subjective and unnecessary.

9. P10 L9-11: Rephrase "The resulting phage amplification rate captures the bactericidal effect independent of initial phage and bacterial population sizes and across different growth conditions" based on your results.

10. P11 L15: "Burst rate" should be changed to "Latent period", as this term is more commonly used to define the time between adsorption to the release of new progeny.

11. P11 L15-17: it is suggested that Fig 1a is based on time-resolved measurement of phage concentrations, but the Y axis reflects bacterial abundance.

12. P11 L27: rephrase "phage amplification rate in dependence of the bacterial concentration" (in dependence is unclear, maybe "depending on")

13. P11 L34: Define BASEL as BActeriophage SElection for your Laboratory.

14. P11 L41 I think the start of the Results section could do with a few introductory sentences. For example, with the term "dynamics of a phage population" do you mean the changes in abundance of the phages over time? I thought you rather "aimed to quantitatively understand" phage virulence parameters? Please clarify. Also on Pg 4 L17.

15. P12 L1: E. coli nor phages are generally bioluminescent. A few more words on the experimental design would be helpful. Additionally, please provide further clarification on how you ensured that all cells were carrying the plasmid throughout the experiment, as plasmid loss over time is known to occur.

16. P12, L5, it is mentioned that the collapse time was measured using bioluminescence. Typically, optical density (OD) is the standard method for such measurements, and the correlation between bioluminescence and OD measurements was initially unclear to this reviewer. I recommend including the current Supplementary Figure 1b within the main Figure 1, as it effectively demonstrates the strong correlation between these two measurement methods. This would provide reassurance about the validity of using bioluminescence as a proxy for optical density.

17. P12 L21-22: please clarify "absolute concentration" of what?

18. P12 L30: fig 2c is mentioned in the text before 2a and 2b, move up in figure, or make a separate figure?

19. P13 L43: the sentence "To put this diversity into perspective..." - please add the actual values, and add some refs.

20. P14 L9: "such as genome size or primary/terminal receptor" where is this data shown? Please refer to a table or figure.

21. P15 L35-36: "phages that showed any interaction at all generally synergized with nitrofurantoin but were antagonized by doxycycline" were these the same phages? A Supplementary Table with all details about the phages in all experiments (not just the antibiotics) would be valuable.

22. P17 L31-32: Please rephrase "phage amplification laws" as it is unclear what you are talking about.

23. P18 L22-26: Can you briefly explain why the spent medium is used for phage dilution, and not fresh medium? Also please describe how the spent medium is made.

24. P18 L34, P19 L14, L18, L20: In the methodology section, the dilution levels of bacteria and phage are mentioned, while in the figures like Fig 2a-b, Fig 3b, the concentrations are provided. To ensure consistency and enhance the clarity of the methodology, could you please include both the dilutions and the corresponding concentrations of bacteria and phage directly in the methodology section.

25. P18 L35: Define ONC as bacterial overnight culture.

26. P19 L38: "Phage purification" From the description this step is to extract phage particles from bacterial culture. At first I thought "phage purification" referred to the method to purify a single phage strain by culturing from a sample, involving multiple plaque assay steps. So maybe an additional context can help to mitigate this? For example "Phage purification from bacterial culture / Phage isolation".

27. P20 L27: "Drop assay for stock concentrations of phage samples" This could be named "Spot assay" to make it easier to understand. The term drop assay is not commonly used.

28. P29 Fig 3c: could you explain where the unit of the X axis comes from, shouldn't this be something like particles per hour?

29. Fig 3d: it looks like Autographiviridae may be significantly higher than the others, is it? Did you correct for multiple testing? This would be appropriate.

30. P30 in Fig 4e you use a P value cutoff of 0.05 ("The dashed line is at p=0.05"). If you want to use a cutoff, it should be corrected for multiple testing.

31. Comparing Fig 4e and 4f, it seems like the axis labels in 4f are reversed? Note that it is difficult to identify the different markers, especially when they overlap. Maybe they can be made a bit bigger, and more different.

32. Fig 4f caption: please put the marker legend in the figure.

33. Several times you mention "bootstrapping" the growth curves, but it is unclear what that means/how this is done. Do you mean replicates? In the fig1 and fig4 captions, could/should you mention how many times (1000x)?

---

## [Decision Letter · Decision Letter 2]

31 Jan 2025

Dear Dr Bollenbach,

Thank you for your patience while we considered your revised manuscript "Extreme diversity of phage amplification rates and phage-antibiotic interactions revealed by PHORCE" for publication as a Short Reports at PLOS Biology. This revised version of your manuscript has been evaluated by the PLOS Biology editors, the Academic Editor and two of the original reviewers.

Based on the reviews, we are likely to accept this manuscript for publication, provided you satisfactorily address the remaining points raised by reviewer 3. Please also make sure to address the following data and other policy-related requests.

a) After discussing within the team and with the Academic Editor, we were thinking that the study could fit well as a Methods and Resources type of article. If you agree that the focus should be on the application of PHORCE, we would encourage to change the type of format during re-submission.

Please supply the numerical values either in the a supplementary file or as a permanent DOI’d deposition for the following figures:

Figure 1bc, 2bd, 3bcde, 4a-e, S1ab, S2ab, S3, S4, S5abc, S6, S7a-d, S8, S9abc, S10abc*

*If some are just examples of the model, please disregard the request

c) Please cite the location of the data clearly in all relevant main and supplementary Figure legends, e.g. “The data underlying this Figure can be found in S1 Data” or “The data underlying this Figure can be found in https://doi.org/10.5281/zenodo.XXXXX”

d) We do not have a word limit, even for Short Reports. Could you please move the section named "Model of phage amplification kinetics" to the Materials and Methods in the main text? This would allow readers an easier access to the model.

e) Please ensure that your Data Statement in the submission system accurately describes where your data can be found and is in final format, as it will be published as written there.

f) Per journal policy, if you have generated any custom code during the course of this investigation, please make it available without restrictions upon publication. Please ensure that the code is sufficiently well documented and reusable, and that your Data Statement in the Editorial Manager submission system accurately describes where your code can be found.

We expect to receive your revised manuscript within two weeks.

*Published Peer Review History*

*Press*

Sincerely,

Melissa

Melissa Vazquez Hernandez, Ph.D.

Associate Editor

PLOS Biology

REVIEWERS' COMMENTS

Reviewer #2:

I apologize for the delays in my review. I am impressed by the revisions that the authors have made, and believe that the new manuscript is suitable for publication in PLoS Biology.

Reviewer #3:

We would like to thank the authors for responding to and addressing most of our previous comments. Overall, this work presents a novel and interesting concept for a new parameter that captures bacteriophage biology, and the effort from the authors to make this protocol accessible to other researchers is evident. However, we still find that the title and abstract suggest that the method is universally applicable, while it is not. We would thus like to offer some further suggestions to be more upfront about the caveats of this work. We feel that this will not detract from the impact, while avoiding potential disappointments for researchers who are not working on fast-growing hosts or adsorption-limited phages. While the addition of Pseudomonas aeruginosa provided valuable insights, it does not address the lack of universal applicability of the PHORCE method. Both E. coli and P. aeruginosa are fast-growing, gram-negative opportunists. To address the issue and move forward, we suggest replacing the sentence in the Abstract: "We found that the resulting phage amplification rate captures the bactericidal effect independent of initial phage and bacterial population sizes and across different growth conditions." by a sentence along the lines of "We found that the resulting phage amplification rate captures the bactericidal effect independent of initial phage and bacterial population sizes, and works well for fast-growing hosts and adsorption-limited phages." Removing the last few words of the original sentence seems justified, since apart from the addition of antibiotics, the study does not address different growth conditions. Please also remove the speculation in line 299 that "differences between, e.g., rich and minimal media or at different temperatures" strongly impact phage amplification, as this seems preliminary.

Re: Reviewer #3 point 2

The authors speculate about the reason that 11/69 E. coli phages do not noticeably affect liquid growth curves, hypothesizing that these phages have such low amplification rates that phage densities do not reach a critical density to cause a collapse before the bacterial culture reaches stationary phase. However, a lack of observable collapse is not always attributed to low amplification rate, but could also reflect a stable coexistence between phage and host (doi: 10.1038/s41467-018-07225-7, 10.1186/s40168-021-01036-7). Please discuss this.

Re: Reviewer #3 point 5

We agree that the observed phenomenon, i.e. the inflection point in the growth curve which is used for calculating the phage amplification rate using PHORCE, most likely stems from phages that are amplifying and killing bacteria. However, there are many examples of situations where phage amplification does not result in a drop in the growth curve (see refs above). In such cases, PHORCE is not suitable for calculating phage amplification. We would like to ask the authors again to reconsider the term "phage amplification rate" to avoid confusion, could they replace by something that includes the killing aspect?

Re: Reviewer #3 point 8

Regarding the word "extreme": the effect spans >3 orders of magnitude (not 4 as stated in the rebuttal). It seems appropriate to be specific, both in the title and Figure 3 caption, so e.g. "Phage amplification rates and phage-antibiotic interactions span three orders of magnitude, as revealed by PHORCE".

The manuscript (and rebuttal) now several times mentions "microscopic phage kinetics", but we find this word confusing, as there is no microscope involved. Please rephrase.

---

## [Editor Report · Decision Letter 3]

12 Feb 2025

Dear Tobias

Thank you for the submission of your revised Methods and Resources "Extreme diversity of phage amplification rates and phage-antibiotic interactions revealed by PHORCE" for publication in PLOS Biology. On behalf of my colleagues and the Academic Editor, Jeremy J. Barr, I am pleased to say that we can in principle accept your manuscript for publication, provided you address any remaining formatting and reporting issues. These will be detailed in an email you should receive within 2-3 business days from our colleagues in the journal operations team; no action is required from you until then. Please note that we will not be able to formally accept your manuscript and schedule it for publication until you have completed any requested changes.

PRESS

Best wishes,

Melissa 

Melissa Vazquez Hernandez, Ph.D., Ph.D.

Associate Editor

PLOS Biology
